# Light promoted brown staining of protoplasm by Ag$^+$ is ideal to test wheat pollen viability rapidly

**Abhishek Biswas◉, Subramaniyan Divya◉, Peddisetty Sharmila, Peddisetty Pardha-Saradhi◉ ***

Department of Environmental Studies, University of Delhi, Delhi, India

◉ These authors contributed equally to this work.
* ppsaradhi@gmail.com

**Data Availability Statement:** All relevant data are within the paper.

**Funding:** The authors received no specific funding for this work.

## Abstract

Pollen viability is crucial for wheat breeding programs. The unique potential of the protoplasm of live cells to turn brown due to the synthesis of silver nanoparticles (AgNPs) through rapid photoreduction of Ag$^+$, was exploited for testing wheat pollen viability. Ag$^+$-viability test medium (consisting of 0.5 mM AgNO$_3$ and 300 mM KNO$_3$) incubated with wheat pollen turned brown within 2 min under intense light (~600 μmol photon flux density m$^{-2}$s$^{-1}$), but not in dark. The brown medium displayed AgNPs-specific surface plasmon resonance band in its absorption spectrum. Light microscopic studies showed the presence of uniformly stained brown protoplasm in viable pollen incubated with Ag$^+$-viability medium in the presence of light. Investigations with transmission electron microscope coupled with energy dispersive X-ray established the presence of distinct 5–35 nm NPs composed of Ag. Powder X-ray diffraction analysis revealed that AgNPs were crystalline and biphasic composed of Ag$^0$ and Ag$_2$O. Conversely, non-viable pollen and heat-killed pollen did not turn brown on incubation with Ag$^+$ medium in light. We believe that the viable wheat pollen turn brown rapidly by bio-transforming Ag$^+$ to AgNPs through photoreduction. Our findings furnish a novel simplest and rapid method for testing wheat pollen viability.

## Introduction

Bread wheat (*Triticum aestivum* L.), the Nature created allohexaploid, is one amongst the three most widely cultivated staple cereal crops. Owing to negative impact of global climate change and linked unpredictable weather conditions and emergence/evolution of newer strains/genotypes of pests and pathogens, crop plants including wheat are exposed to range of abiotic and biotic stresses, resulting in significant decline in potential crop yield [1, 2]. In this era of uncontainable increase in human population and malnutrition, wheat breeders are playing a pivotal role in increasing productivity and enhancing quality of wheat grains through hybridization and biotechnology programs. Wheat pollen, which are shed at 3-celled stage, are short-lived [1]. Pollen viability plays a crucial role in general plant breeding programs, which

**Competing interests:** No The authors have declared that no competing interests exist.

include hybrid seed production, and haploid/double-haploid technology [1–7]. Hence, it is important to monitor pollen viability from time to time. Even though the pollen germination test is considered to be ideal for evaluating viability, researchers often fail to germinate viable pollen due to the requirement of some specific conditions or immaturity. As a result researchers identified alternate protocols that test/detect (i) the dehydrogenase (an enzyme that is omnipresent in all living cells) activity with tetrazolium salts; (ii) callose in pollen walls and tubes with aniline blue; (iii) starch using Lugol solution; (iv) chromatin using Acetocarmine; (v) cytoplasm and cell wall using the Alexander stain; (vi) the esterase activity by means of fluorochromatic reaction; and (vii) the peroxidase activity using phenylenediamine [1–6, 8]. Due to difficulty and lack of consistency in these methods, Heidmann et al. [5] used reliable impedance flow cytometry (IFC) for testing pollen viability. However, IFC test is complicated and expensive as it requires filtration of pollen through specific filters, loading of filtered pollen on specific channel chips, insertion of chips into IFC, and measurement of changes of electrical impedance using special software AmphaSoft v1.2.

In light of limitations in prevailing pollen viability protocols, we attempted to develop novel and more reliable protocol(s) for testing wheat pollen viability. Previously we reported that living cells and chloroplasts have the potential to generate silver nanoparticles as they possess strong reducing strength [9–11]. These findings from our team prompted us to evaluate if wheat pollen viability can be tested based on their potential to generate AgNPs from $Ag^+$. It is known that pollen possess carbohydrates, lipids and proteins besides enzymes like dehydrogenases in their protoplasm [1–5, 12]. These biomolecules can promote reduction $Ag^+$ to generate AgNPs [9, 13–15] and hence can turn protoplasm brown. In this communication we are reporting for the first time that wheat pollen viability can be tested using the potential of their protoplasm to turn brown through photoreduction of $Ag^+$ to form AgNPs.

## Materials and methods

Pollen were collected from bread wheat (*Triticum aestivum* L. cv. 1544, Poaceae), on butter paper from potted plants in our laboratory complex, Department of Environmental Studies, University of Delhi, generally in the morning hours between 8 to 10 AM. $AgNO_3$ and $KNO_3$ were purchased from Merck Specialties Pvt. Ltd. (India). 0.5 mM $AgNO_3$ in combination with 300 mM $KNO_3$ (hereafter referred to as $Ag^+$-viability medium) gave consistent and superior results amongst different concentrations and combinations tested.

Evaluating novel $Ag^+$-medium for testing pollen viability: Within 10–20 min after collection, few pollen grains were incubated in $Ag^+$-viability medium in absence and presence of high-intensity visible light with a photon flux density (PFD) of ~600 µmol $m^{-2}s^{-1}$, for 1 to 5 min under ambient conditions. Subsequently, the pollen grains were observed under Olympus CX40RF200 trinocular microscope (Olympus Optical Co. Ltd., Japan). Images of pollen of all plant species exposed to different treatments were captured using Olympus PEN E-PL1 camera (Olympus Imaging America Inc., USA) attached to the microscope at different magnifications.

For testing if dead/non-viable pollen can also turn brown, we have (i) heat killed the pollen by incubating them in an incubator at 60˚C for 10 min; and (ii) stored them under ambient conditions for 4 h after collection, prior to incubating them in $Ag^+$-viability medium in light.

Characterization of AgNPs: After incubation, the medium with pollen was diluted by adding an equal proportion of distilled water to promote the bursting of pollen. Subsequently this mixture was sonicated at a frequency of 50 kHz for 15 min in an ultrasonic bath (Metrex Scientific Instruments Pvt. Ltd., New Delhi, India) and the absorption spectra of resultant solution was recorded using UV-Vis spectrophotometer (Analytikjena Specord 200, Jena, Germany) at

a resolution of 10 nm and a scan rate of 10 nm/sec. For TEM investigations, 10 µl of the soni-cated mixture was drop coated onto a 200-mesh copper TEM grid with an ultrathin continu-ous carbon film and allowed to dry in a desiccator. The grids were viewed in the transmission electron microscope (Tecnai G2 T30 U-TWIN, Lonate Pozzolo, Italy) at a voltage of 300 kV. The hardware associated with this instrument allowed (a) energy dispersive X-ray (EDX) anal-ysis to measure the elemental composition of NPs; and (b) selected area electron diffraction (SAED) pattern analysis that indicate the crystalline/amorphous nature of NPs. For Powder X-ray diffraction (PXRD) studies the brown sonicated colloidal solution was centrifuged at 10,000 x$g$ and the resultant pellet was suspended in a minimum quantity of double-distilled water. Subsequently, this suspension was drop coated on the silica surface and after drying in a desiccator under ambient conditions, the PXRD pattern was collected using Rigaku Rotaflex RAD-B (Rigaku Analytical Devises, Inc., USA) with copper target CuK($\alpha$)1 radiation at the rate of 0.020 steps in 1.2 s in 2 theta ($\theta$) range of 10–70˚, with a tube voltage of 40 kV and a cur-rent of 60 mA.

All experiments, except for TEM and PXRD analysis, were carried out independently at least six times.

## Results and discussion

Wheat pollen turned $AgNO_3$ solution brown within few seconds on incubation in presence of high-intensity light [~600 µmol PFD m$^{-2}$s$^{-1}$]. Microscopic observations revealed the presence of only burst pollen suspended in the brown colloidal matter. It is well known that pollen burst, when placed in water or any hypotonic solution [16] as the net movement of water into pollen occurs through germ pore(s)/aperture(s), which possess thin wall made of intine with very thin or no exine. Instantaneous diffusion of stained/coloured protoplasm of pollen cells into surrounding hypotonic viability test medium has been recorded by earlier researchers also [16]. To overcome this problem, pollen viability test media are enriched with various com-ponents such as sugars like sucrose and/or salts like $KNO_3$ to enhance their osmotic strength and maintain desired osmotic balance [16]. During present investigations, amongst different components, viz. sucrose, glycerol, $KNO_3$ and KCl tested, 300 mM $KNO_3$ was found to be more suitable for maintaining pollen integrity and staining pollen homogenously brown in presence of 0.5 mM Ag$^+$. Hence, Ag$^+$-viability medium used for present investigations con-sisted of 0.5 mM Ag$^+$ and 300 mM $KNO_3$.

Ag$^+$-viability medium with wheat pollen grains turned brown within 2 min on incubation under ambient conditions in the presence of high-intensity visible light (~600 µmol PFD m$^{-2}$s$^{-1}$), but not in dark (Fig 1A). As evident from Fig 1B, the protoplasm of pollen turned evenly brown on incubation in medium containing Ag$^+$, in presence of high-intensity visible light. It is well established that solution containing Ag$^+$ turns brown due to the generation of AgNPs [9–11, 13–15]. Following incubation of pollen in 300 mM $KNO_3$ containing 0 and 0.5 mM Ag$^+$ in presence and absence of light, the media were diluted with an equal proportion of dis-tilled water to create hypotonic conditions (i.e. osmotic imbalance) that promoted release of pollen content due to bursting, to make the media with pollen relatively homogenous. Absorp-tion spectra of sonicated resultant pale yellow (control) or brown (Ag$^+$-treated) colloidal sus-pensions of wheat pollen incubated in presence of light are depicted in Fig 1C. Brown colloidal suspension obtained from the Ag$^+$ viability medium incubated with wheat pollen in light showed a prominent peak around ~420 nm (Fig 1B). It is well known that this peak around ~420 nm arises due to surface plasmon resonance specific to AgNPs [9–11], which ascertained the potential of wheat pollen to generate AgNPs rapidly on incubation in Ag$^+$-viability medium in presence of high-intensity light.

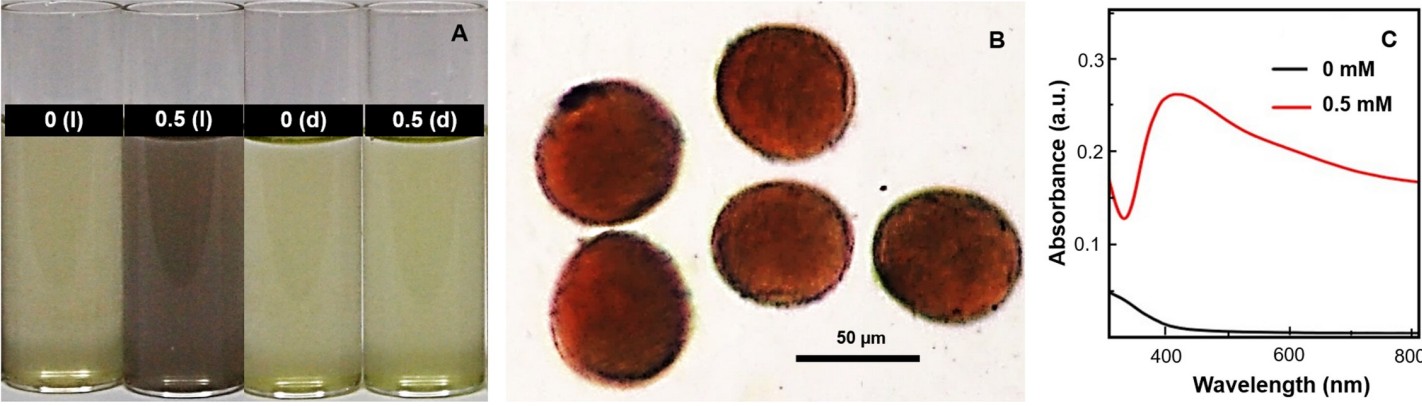

**Fig 1. Potential of Ag⁺-viability medium (consisting of 0.5 mM AgNO₃ and 300 mM KNO₃) to stain wheat pollen brown in presence of high-intensity light (~600 μmol PFD m⁻²s⁻¹).** (A) Variation in color of medium (300 mM KNO₃) with 0 and 0.5 mM Ag⁺ incubated with wheat pollen in presence of high-intensity visible light (l) and dark (d) for 5 min; (B) Light microscopic image of wheat pollen showing brown stain; (C) Absorption spectra of sonicated media with wheat pollen incubated with 0 and 0.5 mM Ag⁺ in the presence of high-intensity visible light.

TEM investigations confirmed the presence of distinct 5 to 35 nm nanoparticles in brown solutions formed on incubation of pollen in Ag⁺-viability medium in the presence of light (Fig 2A). The energy dispersive X-ray (EDX) microanalysis pattern showed distinct peaks around 3.40 keV (Fig 2B), that correspond to Ag [14], ascertaining the presence of Ag in these NPs. The peaks of C and Cu arose due to their presence as a basic component of carbon coated copper grids, while that of O arose due to its presence in biphasic Ag⁰/Ag₂ONPs. Selected area electron diffraction (SAED) pattern displayed the presence of distinct rings corresponding to Bragg reflections (Fig 2C), revealing that these AgNPs were crystalline [14].

The powder X-ray diffraction (PXRD) pattern of the brown colloidal matter that resulted due to incubation of wheat pollen in Ag⁺-viability medium in presence of intense light (Fig 2D) showed Bragg reflections at (a) (111), (200) and (311), which matched with the standard diffraction pattern of Joint Committee on Powder Diffraction Standards (JCPDS) No. 89–3722, indicating the face-centered cubic (fcc) structure of Ag⁰; and (b) (111)*, (211)*, (220)* and (221)* that matched with JCPDS No. 76–1393 characteristic of Ag₂O with cubic geometry [14]. The presence of distinct peaks (Fig 2D) further corroborated the crystalline nature of AgNPs [14]. These PXRD results revealed that the AgNPs formed by wheat pollen were biphasic composed of Ag⁰ and Ag₂O. It is well-known that under ambient aerobic conditions, Ag⁰ and Ag⁰NPs are prone to oxidation, which results in the generation of Ag₂ONPs [9–11, 14].

Previously, we reported the potential of (i) live leaf cells; and (ii) chloroplasts to reduce Ag⁺ to generate AgNPs in presence of high-intensity light (~600 μmol PFD m⁻² s⁻¹) under ambient conditions [10, 11]. As evident from Fig 3A, over 60 percent of wheat pollen stored for 4 h under ambient conditions failed to turn brown on incubation with Ag⁺-viability medium in presence of light, revealing loss in viability on storage. Wheat pollen are short-lived as they are shed at 3-celled stage [1]. Heat killed wheat pollen incubated in Ag⁺-viability medium in presence of high-intensity light failed to turn brown (Fig 3B), revealing the necessity of live protoplasm within pollen for the rapid brown staining through generation of AgNPs.

Mechanism of brown staining of viable wheat pollen through the generation of AgNPs: Wheat pollen, are enriched with a range of biomolecules, which include sugars like sucrose, glucose, fructose, and raffinose besides amino acids, proteins etc. for generating desired energy, redox power and range of carbon skeletons. Several such biomolecules promote photoreduction and biotransformation of Ag⁺ to AgNPs [10, 17, 18]. Guo and co-workers [18]

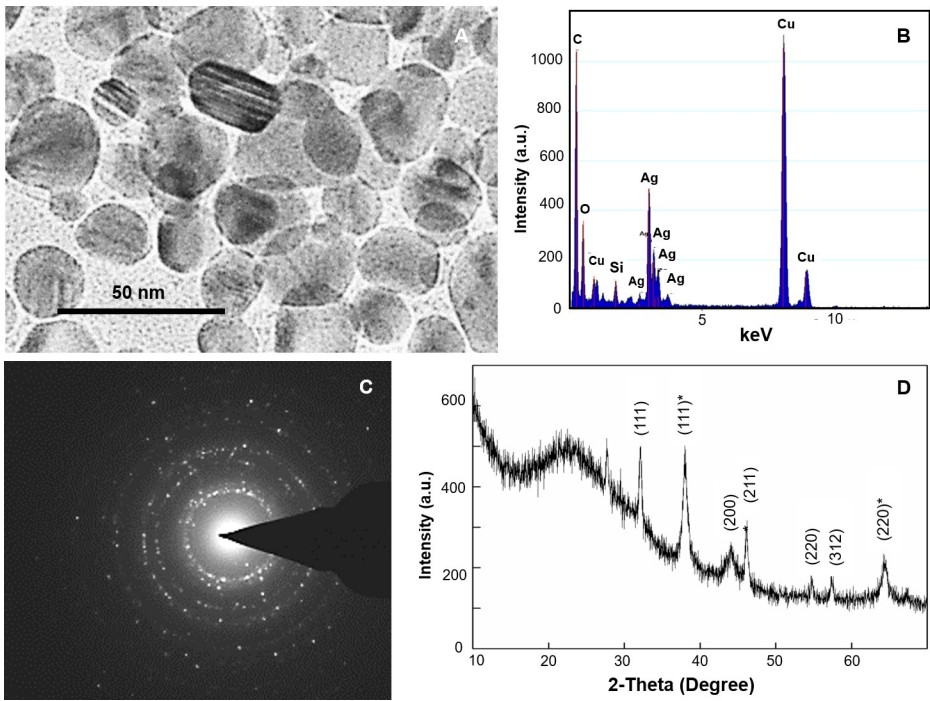

**Fig 2. Ag$^+$-viability medium imparted brown staining of wheat pollen is due to AgNPs.** (A) Transmission electron micrograph show distinct AgNPs in the sonicated pollen after incubation with Ag$^+$-viability medium in presence of light; (B) Energy dispersive X-ray (EDX) pattern showing Ag peaks; (C) Selected area electron diffraction (SAED) pattern; (D) PXRD pattern of AgNPs formed by the wheat pollen incubated with Ag$^+$ viability medium in presence of light. Bragg reflections confirmed that these AgNPs are biphasic composed of Ag$^0$ '()' and Ag$_2$O '()*'.

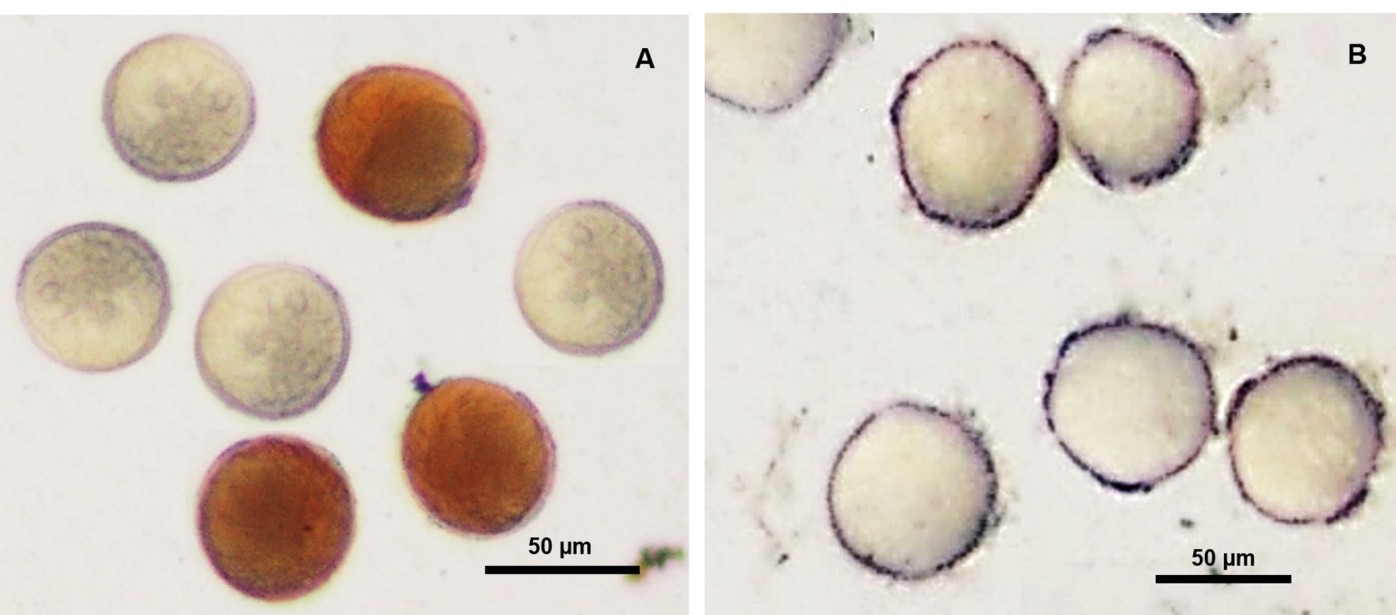

**Fig 3.** Inability of (A) wheat pollen that turned non-viable on storage for 4 h; and (B) heat-killed wheat pollen to stain brown on incubation with Ag$^+$-pollen viability medium in presence of light, indicating that live protoplasm is essential for rapid photoreduction of Ag$^+$ for uniform brown staining. Please note mixture of viable and non-viable pollen in (A).

recorded sunlight promoted biotransformation of Ag$^+$ into AgNPs by biomolecules from root exudates. Further, we have earlier reported that dehydrogenases promote reduction Ag$^+$ to generate AgNPs [9] and pollen possesses dehydrogenases [1–5, 16]. It is well known that Ag$^0$ and Ag$^0$NPs are prone to oxidation under aerobic conditions [9–11] and accordingly, as evident from PXRD analysis (Fig 2D) AgNPs generated by pollen through photoreduction are biphasic made of Ag$^0$ and Ag$_2$O. Therefore, we strongly believe that viable pollen, which invariably exhibits active metabolism, generates biphasic AgNPs through photoreduction using inbuilt biomolecules and dehydrogenases. These biphasic AgNPs turn the protoplasm of viable pollen cells brown and hence Ag$^+$-viability medium can be used authentically to rapidly test viability of wheat pollen.

In summary, we developed an authentic protocol for testing wheat pollen viability, which involved (i) incubation of wheat pollen in Ag$^+$-viability medium (consisting of 0.5 mM AgNO$_3$ and 300 mM KNO$_3$) in presence of high-intensity light (~600 μmol PFD m$^{-2}$s$^{-1}$) for 2 min under ambient conditions; and (ii) light microscopic recording of brown staining of viable pollen. Ag$^+$-pollen viability test failed to stain stored non-viable and heat-killed pollen brown, indicating that live protoplasm is essential for rapid photoreduction and biotransformation of Ag$^+$ to AgNPs. We believe that this Ag$^+$-pollen viability test can be extended to other crop plants. To the best of our knowledge, the novel Ag$^+$-wheat pollen viability test developed by us is the simplest rapid, economically viable, and green method amongst the prevailing pollen viability tests.

## Author Contributions

**Conceptualization:** Peddisetty Sharmila, Peddisetty Pardha-Saradhi.

**Formal analysis:** Peddisetty Pardha-Saradhi.

**Funding acquisition:** Peddisetty Pardha-Saradhi.

**Investigation:** Abhishek Biswas, Subramaniyan Divya, Peddisetty Sharmila, Peddisetty Pardha-Saradhi.

**Methodology:** Peddisetty Pardha-Saradhi.

**Project administration:** Peddisetty Pardha-Saradhi.

**Resources:** Peddisetty Sharmila, Peddisetty Pardha-Saradhi.

**Supervision:** Peddisetty Pardha-Saradhi.

**Validation:** Peddisetty Pardha-Saradhi.

**Writing – original draft:** Abhishek Biswas, Subramaniyan Divya, Peddisetty Pardha-Saradhi.

**Writing – review & editing:** Subramaniyan Divya, Peddisetty Sharmila, Peddisetty Pardha-Saradhi.

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
