## [Decision Letter · Decision Letter 0]

11 Jun 2020

PONE-D-20-16107

Potential to generate Ag-nanoparticles can be used as an ideal tool to determine pollen viability

PLOS ONE

Dear Dr. Pardha-Saradhi,

Thank you for submitting your manuscript to PLOS ONE. After careful consideration, we have decided that your manuscript does not meet our criteria for publication and must therefore be rejected.

I am sorry that we cannot be more positive on this occasion, but hope that you appreciate the reasons for this decision.

Yours sincerely,

Vijay Kumar

Academic Editor

PLOS ONE

Reviewers' comments:

Reviewer's Responses to Questions

**Comments to the Author**

1. Is the manuscript technically sound, and do the data support the conclusions?

Reviewer #1: Yes

Reviewer #2: No

2. Has the statistical analysis been performed appropriately and rigorously? 

Reviewer #1: No

Reviewer #2: N/A

3. Have the authors made all data underlying the findings in their manuscript fully available?

Reviewer #1: Yes

Reviewer #2: Yes

4. Is the manuscript presented in an intelligible fashion and written in standard English?

Reviewer #1: Yes

Reviewer #2: Yes

5. Review Comments to the Author

Reviewer #1: The manuscript entitled “Potential to generate Ag-nanoparticles can be used as an ideal tool to determine pollen viability” has been reviewed. It is very general observation that plant cells and extracts could be used for biogenic synthesis of AgNPs. The manuscript is written well and proper justifications are given to the results obtained. There are certain points to be rectified before further processing of the manuscript:

• What was the criterion in selection of plants for pollen viability testing?

• Why these plants only selected?

• Part of second paragraph of Material and methods section should be presented in Results and discussion section.

• Even some parts of R and D section can be part of M and M section too.

• What was the standard solution used for the spectroscopic analysis of AgNP containing pollens?

• Write in details about the data analysis of this study.

• “Ag+ viability medium with wheat pollen grains turned brown within 2 min on incubation under ambient conditions in presence of solar radiation or high intensity visible light, but not in dark (Fig 1A). As evident from Fig 1B, protoplasm of pollen turned evenly brown on incubation in medium containing Ag+, only in presence of solar radiation or high intensity visible light. It is well established that solution containing Ag+ turns brown due to generation of AgNPs”....... The authors must discuss in details about the role of light in formation of AgNPs. Because, they got AgNPs in light conditions only.

• Sentence should not be started with the abbreviations....

• It is well established fact that living cells and cellular organelles help in biogenesis of AgNPs, therefore, the pollen too helped in this study. Weather such pollens could be used for plant breeding experiments without losing their viability after used in this experiment?

• “in presence of solar radiation or high intensity light (Fig 1F & Fig 2L-2O). These findings, further establish that active/live protoplasm is necessary to turn pollen brown due to potential to generate AgNPs”.... But there are reports which used plant extracts only to generate AgNPs in the solution within one min.....any clarification in this regard...

• Please check “miccrocubes”

• References formatting must be uniform based on journal’s requirement.

• Authors should provide page and line numbers while revising the manuscript.

Reviewer #2: The manuscript has some serious issues that need to be addressed:

1. Experiment with a non-biological control (no pollen) wherein only Silver salt and KCl in the presence of light should be conducted.

2. Experiment with a non-biological control (no pollen) wherein only Silver salt and KCl in the absence of light should also be conducted. Please refer to the paper wherein SNPs (AgNPs) and GNPs were generated with heat-killed date palm pollens in the dark.

3. There are ample reports describing synthesis of SNPs using whole cell, cell free extract, plant extract for synthesis of nanoparticles using Ag salts only. I would like to know why KCl was used in the reaction as it is well known that Ag salt will be precipitated as Ag chloride.

4. Experiment using Ag salts and pollen may also be conducted.

5. Authors have not divulged details on various accounts, e.g. sonication conditions, heat-killing treatment and aged pollen.

6. Please see the annotated pdf for detailed comments.

6. PLOS authors have the option to publish the peer review history of their article (what does this mean?). If published, this will include your full peer review and any attached files.

Reviewer #1: No

Reviewer #2: No

- - - - -

---

## [Author Response · Author response to Decision Letter 0]

13 Aug 2020

Academic Editor’s Comment: “After careful consideration, we have decided that your manuscript does not meet our criteria for publication and must therefore be rejected.”

Response: According to both the Reviewers 1 and 2, the manuscript was presented in an intelligible fashion and written in standard English. Reviewer 1 further stated that “The manuscript is written well and proper justifications are given to the results obtained.”

Further, all the issues raised by the reviewer are minor in nature and can be rectified with ease. To the best of our knowledge, none of the issues raised by the Reviewers are serious enough for “Rejection”. 

Unfortunately, both the Academic Editor (Dr. Vijay Kumar) as well as Reviewers failed to understand the basic goal/objective of our manuscript. All of them seem to be experts, and interested and accordingly focused on outdated biogenic synthesis of Ag-nanoparticles (especially with extracts of plant parts including pollen). It is surprising that the Academic Editor, Dr. Vijay Kumar failed to give his own independent comments/views [he used only standard Rejection letter of PLoS One, without furnishing any Specific comments for the Rejection]. As none of the comments of the Reviewers are serious in nature, we believe that the “Rejection” decision taken by the Academic Editor, Dr. Vijay Kumar is 100% biased and it only reflects that he has some personal vengeance against us. 

By looking at vagueness of the decision and comments, I initially preferred not to be part of PLoS One community any further. But the co-authors insisted me to Appeal to expose biased attitude of the Academic Editor, Dr. Vijay Kumar, to curb such illogical decisions. However, while respecting the comments [albeit many of them are vague], we annotated the manuscript significantly. 

As elaborating on the reasons for using each plant/crop species, would unnecessarily (i) dilute the key objective and finding of our investigations; and (ii) increase the length of the manuscript, we decided to restrict to only wheat [as detailed investigations were carried with this plant species] and also furnish data obtained using Ag+-viability medium consisting of AgNO3 and KNO3 [and totally omitted the information related with KCl]. In fact, the entire data furnished in the original manuscript was obtained using Ag+-viability medium consisting of AgNO3 and KNO3. Although, we obtained similar results even when we replaced KNO3 with KCl [as both these salts are used as osmotic agents to prevent pollen bursting]. The same has been stated in our original manuscript. Unfortunately, due to negligence we mistakenly typed KCl in place of KNO3 while disclosing the composition of Ag+-viability medium. This mistake was committed as we wanted to disclose interesting observation, wherein we recorded staining of pollen brown by AgNO3 even in presence of KCl, although AgNO3+ K Cl mixture turned colloidal due to formation of AgCl [we indeed specifically pointed out the formation of colloidal medium due to the formation of AgCl in our Original manuscript]. Probably the Academic Editor as well as the Reviewer 2 failed to go through our manuscript judiciously. To avoid any confusion, now we deleted KCl results. Please find below our comments on every point raised by both the Reviewers.

Reviewer #1 (R1)

R1 Comment: The manuscript entitled “Potential to generate Ag-nanoparticles can be used as an ideal tool to determine pollen viability” has been reviewed. “It is very general observation that plant cells and extracts could be used for biogenic synthesis of AgNPs.” 

“The manuscript is written well and proper justifications are given to the results obtained.”

Response: This Reviewer specifically stated/agreed that the manuscript is written well and proper justifications are given to the results obtained. We fail to understand how can a manuscript that is written well with proper justification to the findings/results can be blindly rejected by the Academic Editor Dr. Vijay Kumar. We accept that the biogenic synthesis of AgNPs by plant cells and extracts is a very general observation. However, objective of our investigations presented in the manuscript is not the general biogenic synthesis of AgNPs. Our aim was to develop a simple method for testing pollen viability. We used Ag+ for this endeavor and specifically demonstrated that the pollen viability can be tested using AgNO3. However, to explain the mechanism associated with brown staining, we specified that brown staining of pollen on incubation with AgNO3 was due to synthesis of AgNPs [which have been characterized for proving the mechanism associated with staining)]. 

R1 Comment: There are certain points to be rectified before further processing of the manuscript:

“• What was the criterion in selection of plants for pollen viability testing?”

“• Why these plants only selected?”

Response: Wheat is the major cereal crop [this information is now incorporated in the annotated manuscript] and continuous efforts are being made across the globe for its improvement through plant breeding program. Viable pollen is the key for successful plant breeding. 

Similarly, pigeon pea is an important pulse crop, Indian mustard is an important oil crop fennel is a spice crop, hemp is an important fiber crop, onion is an important vegetable crop. Black night shade is an important medicinal plant. Others are important floricultural crops, each of which possess some medicinal utility besides aesthetic value. Most importantly, all of them were easily available for our studies without any barrier. We don’t believe in increasing manuscript size by describing purpose of using each of them by briefing on importance of each floricultural plant species. In fact, such efforts will dilute the key objective of any useful investigation. We strongly believe that the manuscript should be crisp and focus on findings related to the objective/hypothesis, rather than unnecessarily increasing the length of the manuscript. Therefore, in annotated manuscript, we restricted ourselves to only wheat as we carried detailed and complete investigations with wheat pollen.

In fact, we submitted our manuscript as a Brief Communication to PLoS Biology and the same has been transferred to PLoS One [as an internal policy, after seeking permission of the Authors].

R1 Comment: “• Part of second paragraph of Material and methods section should be presented in Results and discussion section.”

“• Even some parts of R and D section can be part of M and M section too.”

Response: Necessary action taken in Annotated manuscript.

R1 Comment: “• What was the standard solution used for the spectroscopic analysis of AgNP containing pollens?”

Response: We used mixture containing 0.5 mM AgNO3 and 300 mM KNO3, which has been diluted by adding equal volume of distilled water, in a manner similar to the way viability medium incubated with pollen was diluted. The later has been stated clearly in our manuscript.

R1 Comment: “• Write in details about the data analysis of this study.”

Response: We have already furnished information on data analysis. It is not clear what additional ‘details about the data analysis’ are required. However, we have further elaborated Methods.

R1 Comment: “• Ag+ viability medium with wheat pollen grains turned brown within 2 min on incubation under ambient conditions in presence of solar radiation or high intensity visible light, but not in dark (Fig 1A). As evident from Fig 1B, protoplasm of pollen turned evenly brown on incubation in medium containing Ag+, only in presence of solar radiation or high intensity visible light. It is well established that solution containing Ag+ turns brown due to generation of AgNPs”....... The authors must discuss in details about the role of light in formation of AgNPs. Because, they got AgNPs in light conditions only.”

Response: At this stage, it is difficult for us to discuss details about the actual role of light in formation of AgNPs. We stated that high intensity light promoted rapid generation of AgNPs. However, if pollen are incubated under lab conditions for long duration, one can still see alteration in colour of viability medium with pollen, albeit the intensity of brown colour is low. Our objective is to develop a rapid pollen viability test. It is important to again stress that our aim was not to generate AgNPs. Generation of AgNPs by plant extracts has been carried out by innumerable number of investigators (surprisingly, even today such works get published) and in fact presently, generation of AgNPs by plant extracts has become a simple lab experiment for school and college students. At present, such works certainly don’t make any sense. 

R1 Comment: “• Sentence should not be started with the abbreviations....”

Response: Suggested corrections incorporated.

R1 Comment:”• It is well established fact that living cells and cellular organelles help in biogenesis of AgNPs, therefore, the pollen too helped in this study. Weather such pollens could be used for plant breeding experiments without losing their viability after used in this experiment?”

Response: To the best of our knowledge, use of silver salts for testing viability of pollen is novel. Yes, besides few other investigators, our research team has also authentically demonstrated that live cells as well as live chloroplasts possess potential to generate Ag-nanoparticles [some of which have been published in PLoS One itself]. In fact, this is the basic background (hypothesis) based on which we attempted to test if such a potential of live cells to generate Ag-nanoparticles can be exploited for testing pollen viability. It is relevant to cite few classical examples related for testing pollen viability itself. Earlier, few dynamic and pioneering researchers used TTC assay, esterase activity, peroxidase activity, IKI starch staining, acetocarmine test etc., all of which are known to exist in live cells for several decades. But, these dynamic researchers got due credit for using such approaches for testing pollen viability. We also wish to quote a dynamic pollen viability test published in PLOS One itself [Heidmann I, Schade-Kampmann G, Lambalk J, Ottiger M, Di Berardino M. Impedance flow cytometry: a novel technique in pollen analysis. PLoS One 2016;11: 0165531 doi: 10.1371/journal.pone.0165531], wherein Impedance flow cytometry was used for testing pollen viability. In fact, due to importance and relevance, researchers across the globe have been making efforts in developing novel and simple protocols for testing pollen viability. Similarly, recently Impe & co-workers published a paper on assessment of pollen viability in wheat in Frontiers in Plant science [Impe D, Reitz J, C Köpnick, Rolletschek H, Börner A, Senula A, Nagel M. Assessment of pollen viability for wheat. Front Plant Sci. 2020;10: 1588 doi: 10.3389/fpls.2019.01588]. Compared to the protocols used by these investigators as well as others so far, the pollen viability test developed by us is rapid, simplest and cost effective. 

R1 Comment: “• “in presence of solar radiation or high intensity light (Fig 1F & Fig 2L-2O). These findings, further establish that active/live protoplasm is necessary to turn pollen brown due to potential to generate AgNPs”.... But there are reports which used plant extracts only to generate AgNPs in the solution within one min.....any clarification in this regard...”

Response: It looks that this reviewer failed to understand/grasp the basic hypothesis/objective of our investigations presented in this manuscript. Our aim was certainly not to furnish a protocol for generation of Ag-nanoparticles. Use of plant extracts for generation of Ag-nanoparticles has been known for several decades and presently, it is well known that such findings using blind use of extract of different parts of various plant species has no relevance as plant extracts possess innumerable types of reducing agents [such as phenolics, ascorbic acid, amino acids, organic acids, sugars, starch etc. etc.]. Accordingly, one gets highly heterogenous Ag-nanoparticles. If use of plants extracts has any significance, the same could have been used for commercial generation of Ag-nanoparticles. The same has been referred in our previous publications [including those appeared in PLoS One]. We could have given more specific comments, if this knowledgeable Reviewer could have pointed specific publications.

R1 Comment: “• Please check “miccrocubes””

Response: Our aim is simple [i.e. furnishing a rapid and simple protocol for testing pollen viability]. We don’t find any relevance for testing “microcubes” [we don’t believe in blindly copying, what others have reported, until and unless it has relevance].

R1 Comment: “• References formatting must be uniform based on journal’s requirement.”

Response: This manuscript was initially submitted to PLoS Biology and the same has been transferred directly from PLos Biology website to PLoS One website. Yes, references have now been formatted in accordance with PLoS One’s requirement.

R1 Comment: “• Authors should provide page and line numbers while revising the manuscript.”

Response: Page and line numbers have now been included.

Reviewer #2 

R2 Comment: “The manuscript has some serious issues that need to be addressed:”

Response: This Reviewer 2 stated that there are some serious issues that need to be addressed. If the issues raised by this Reviewer can be addressed, how can the Academic Editor take a decision to reject our manuscript, without furnishing specific/valid reasons. Each of the “some serious issues” raised by this Reviewer is addressed below.

R2 Comment:1. Experiment with a non-biological control (no pollen) wherein only Silver salt and KCl in the presence of light should be conducted.

Response: This was very much carried out, but inclusion of this information doesn’t add any significant addition to the manuscript. Further in annotated manuscript we deleted our findings with KCl.

R2 Comment:2. Experiment with a non-biological control (no pollen) wherein only Silver salt and KCl in the absence of light should also be conducted. Please refer to the paper wherein SNPs (AgNPs) and GNPs were generated with heat-killed date palm pollens in the dark.

Response: This was also very much carried out and we did categorically state in our manuscript that “on mixing AgNO3 with KCl, an opaque white colloidal solution was produced as a result of formation of AgCl within a short duration”. Therefore, we used KNO3 and not KCl, although we do get similar results with KCl. The same has been stated in Materials and methods. Unfortunately, we have mistakenly referred KNO3 as KCl. We apologize for the same. 

We have carefully gone through the paper of Banu et al. (2018) referred by this reviewer and noted that (i) heat-killed date palm pollen were not directly used for generation of Ag and Au nanoparticles; (ii) the extract obtained by heating stored pollen dust, was used for generating nanoparticles in dark; and (iii) investigations were carried with the presumption that the extract contains a range of phytochemicals. No attempts were made by these authors to evaluate if pollen themselves can generate nanoparticles. Hundreds of papers have been published where researchers demonstrated potential of extracts of plant parts of various plant species to generate nanoparticles. We don’t see any reason, why we need to cite this paper as (i) pollen were not directly used for generation of nanoparticles; and (ii) the authors showed photos of dried inflorescence of date palm as a figure and not of pollen. Further, the findings presented in Banu et al. (2018) are nothing to do with pollen viability or any way closure to the hypothesis based on which we performed our investigations presented in our manuscript. As stated previously, we wish to again stress that our aim was not to generate Ag-nanoparticles. Unfortunately, this expert reviewer seems to be interested only in generation of Ag-nanoparticles, that too using extracts of plants parts. Our research team never believed in blindly using extracts [which contain wide range of biomolecules with different degrees of reducing capacity] for generating metal nanoparticles.

R2 Comment: “3. There are ample reports describing synthesis of SNPs using whole cell, cell free extract, plant extract for synthesis of nanoparticles using Ag salts only. I would like to know why KCl was used in the reaction as it is well known that Ag salt will be precipitated as Ag chloride.”

Response: This comment clearly reflects that this Reviewer as well as the Academic Editor Dr. Vijay Kumar have gone through our manuscript very casually, overlooking the key goal of the manuscript and contents elaborated in our manuscript. Our research team itself has reported potential of chloroplasts and root cell (in particular) epidermal cells generate Ag-nanoparticles. Use of plant extracts to generate AgNPs is known from last three decades. Objective of our investigations is not to develop any AgNPs generating protocol. We specifically stated that our objective was to test if rapid AgNPs synthesis capacity of live pollen can be exploited for testing pollen viability.

In our original manuscript, we had categorically stated that “on mixing AgNO3 with KCl, an opaque white colloidal solution was produced as a result of formation of AgCl within a short duration”. This itself reflects that we are very well aware that Ag+ precipitates as AgCl. That is the precise reason, why we stated in Materials and Methods “Therefore, it is important to disclose specifically that 1 mM AgNO3 and 600 mM KCl solutions must be stored separately and former solution be added soon after pollen were incubated in later solution.” Although, we were getting ideal results with KNO3 (which we used for all our investigatiosn presented in the manuscript), we gave this information as we were getting equally good results with KCl. However, through the above referred statement, we wanted researchers (who prefer using KCl) to know that they should incubate pollen first in KCl and then add AgNO3. This allows K+ and Cl- to be absorbed by the pollen grains before Ag+. Moreover, the concentration of Ag+ used was 600 times lower than that of KCl. Further, it is important to disclose true facts to the scientific community. If we wanted, we could have avoided disclosing results we obtained with KCl. To avoid any sort of confusion, now we deleted the findings with KCl in annotated manuscript.

R2 Comment: “4. Experiment using Ag salts and pollen may also be conducted.”

Response: It looks that due to casualness and lack of desired knowledge of pollen biology, this comment was made. We have specifically stated in our manuscript [based on authentic reports from several researchers, besides common observation made by even school kids studying pollen germination/biology], the significance of using an osmotic component to prevent bursting of pollen. We also stated categorically that, we diluted viability medium with pollen incubated in it, to burst the pollen so that the contents get released due to bursting. Therefore, performing any investigations of pollen viability just with silver salt is unscientific and senseless effort as live pollen burst due to severe osmotic shock (extreme hypotonic condition). However, if someone is interested in generating Ag-nanoparticles, such an effort would be ideal. It looks that both this reviewer and the Academic Editor, Dr. Vijay Kumar seems to be experts and only interested in generation of AgNPs and not concerned about the key objective of testing pollen viability. We feel extremely sorry to state that the Academic Editor Dr. Vijay Kumar and this Reviewer should have not taken up this assignment of evaluating our manuscript as they lacked basic background of pollen biology. Both of them seems to be probably experts in generation of AgNPs using plant extracts orelse they are extremely biased due to reasons best known to them.

R2 Comment: “5. Authors have not divulged details on various accounts, e.g. sonication conditions, heat-killing treatment and aged pollen.”

Response: As stated previously, we submitted this manuscript to PLoS Biology as a Brief Communication and hence didn’t elaborate on “sonication conditions, heat-killing treatment and aged pollen”. Now, we included the same in our annotated manuscript.

R2 Comment:6. Please see the annotated pdf for detailed comments.

Suggestion of the Reviewer 2: “Subsequent to pollen rupturing” instead of “Subsequent to rupturing pollen”.

Response: We don’t accept this suggestion as it gives a wrong impression. It only reflects that the Reviewer 2 seems to be not competent enough to understand the basics of pollen biology and the objective of our manuscript.

In addition, more or less similar comments as above were marked on the PDF. The same have been extracted in JPG format and pasted below along with our specific response:

R2 Comment on PDF:

Response: It is well known/established fact that viability of pollen varies from species to species. Pollen tend to lose viability on storage, especially under ambient conditions. The term ”aged” was used by us for the pollen stored under ambient conditions for different time intervals. To avoid confusion, now we deleted the term “aged” and stated the duration of storage in the annotated manuscript. 

R2 Comment on PDF:

Response: These comments are irrelevant and only reflects not only biased attitude as this reviewer seems to have some personal problem with us or he/she has some unknown personal gain through rejection. I don’t see any valid reason why we need to unnecessarily add a Table to increase the length of the manuscript. There is absolutely nothing wrong in simply listing the plant species used. However, authority name has now been included. It is irrelevant to refer to www.ipni.org as there are superior sites. 

The question of “where are the vouchers of the species used deposited?”, only reflects negative mindset of the Reviewer [as well as the Academic Editor Dr. Vijay Kumar”, who entertained such incompetent comments]. Can Dr. Vijay Kumar let me know what sort of vouchers this Reviewer is referring to? Does every author who is using a plant species authorized to submit such vouchers? 

With regard to the reason, why we selected different plant species, we made it crystal clear in our original manuscript that “In order to check, if Ag+ viability medium can be used for testing pollen viability in all plant species, we extended our studies to 20 plant species …..” besides their availability. We are afraid that this Reviewer is either biased or doesn’t understand simple language. It looks that both this Reviewer as well as Academic Editor believe in promoting lengthy manuscripts filled with useless packing material, rather than having crisp manuscript. 

R2 Comment on PDF:

Response: That is the precise reason, why we categorically stated the same in our manuscript. We disagree that it is essential to use SEM, AFM to confirm the presence of AgNPs. I am shocked. It is not clear how AFM confirms the presence of AgNPs. To the best of our knowledge Transmission electron microscope coupled with the hardware for (a) energy dispersive X-ray (EDX) analysis; and (b) selected area electron diffraction (SAED) pattern analysis, is good enough to characterize AgNPs. In addition, we carried out vital PXRD analysis. 

R2 Comment on PDF:

Response: This statement of the Reviewer, clearly reflects the way he/she can mislead the scientific world. We have carefully gone through the paper of Banu et al. (2018) referred by this reviewer and noted that (i) heat-killed date palm pollen were not directly used for generation of Ag and Au nanoparticles; (ii) the extract obtained by heating stored pollen dust, was used for generating nanoparticles in dark; and (iii) investigations were carried with the presumption that the extract contains a range of phytochemicals. No attempts were made by these authors to evaluate if pollen themselves can generate nanoparticles. The authors showed photos of dried inflorescence of date palm as a figure and not of pollen. Further, the findings presented in Banu et al. (2018) are nothing to do with pollen viability or any way closure to the hypothesis based on which we performed our investigations presented in our manuscript. As stated previously, we wish to again stress that our aim was not to generate Ag-nanoparticles. Unfortunately, this expert reviewer seems to be interested only in generation of Ag-nanoparticles, that too using extracts of plants parts. Our research team never believed in blindly using extracts [which contain wide range of biomolecules with different degrees of reducing capacity] for generating metal nanoparticles.

R2 Comment on PDF:

Original: “grasped”

Reviewer 2’s suggestion: “captured”

Response: This suggestion has been incorporated in the annotated manuscript..

R2 Comment on PDF:

Response: Sonication conditions have been included in the annotated manuscript.

R2 Comment on PDF:

Response: These statements again reflects that this Reviewer has not gone through our manuscript carefully. He could not grasp basic objective, hypothesis and the novel findings of the investigations presented in our manuscript. 

R2 Comment on PDF:

Response: Details of how pollen was heat killed has now been elaborated in the annotated manuscript.

R2 Comment on PDF: Names of all Journals have been highlighted as they were represented in full form rather than in abbreviated form. 

Response: Finally, Reviewer 2 given his comment against name of every journal in Reference section, as full name of every journal was given instead of in abbreviated form. As stated previously, our manuscript was transferred directly from PLoS Biology website to PLoS One website as per the accepted terms and conditions. Therefore, the format of references was in PLoS Biology style. Names of Journals can easily be abbreviated. This can’t be taken as a criterion for rejection of any manuscript. 

We have now abbreviated the names of the journals in the annotated manuscript.

---

## [Decision Letter · Decision Letter 1]

4 Nov 2020

PONE-D-20-16107R1

Light promoted brown staining of protoplasm by Ag+ is ideal for testing wheat pollen viability

PLOS ONE

Dear Dr. Pardha-Saradhi,

Thank you for submitting your manuscript to PLOS ONE. After careful consideration, we feel that it has merit but does not fully meet PLOS ONE’s publication criteria as it currently stands. Therefore, we invite you to submit a revised version of the manuscript that addresses the points raised during the review process.

It looks OK, but I have a few comments that need to be addressed by the authors before I can recommend the paper to be published.

1.) The author needs to revise the title.

2.) The data are mostly qualitative, therefore the authors need to tone down their claims about the novelty of the methodology.

3.) The authors need to carefully address the Reviewers comments in the manuscript (responding to the comments is not sufficient).

4.) They authors should add more data to the paper. I edited many papers for PLoS One, but I never approved a paper with only one figure with mostly qualitative data. 

We look forward to receiving your revised manuscript.

Kind regards,

Zhong-Hua Chen, Ph.D.

Rohit Joshi, Ph.D.

Academic Editors

PLOS ONE

Journal Requirements:

2. Thank you for stating the following financial disclosure: 'No

The funders had no role in study design, data collection and analysis, decision to publish, or preparation of the manuscript.'At this time, please address the following queries:

d) If you did not receive any funding for this study, please state: “The authors received no specific funding for this work.”Please include your amended statements within your cover letter; we will change the online submission form on your behalf.

3. Please remove your figures/ from within your manuscript file, leaving only the individual TIFF/EPS image files. These will be automatically included in the reviewer’s PDF

Reviewers' comments:

Reviewer's Responses to Questions

**Comments to the Author**

1. If the authors have adequately addressed your comments raised in a previous round of review and you feel that this manuscript is now acceptable for publication, you may indicate that here to bypass the “Comments to the Author” section, enter your conflict of interest statement in the “Confidential to Editor” section, and submit your "Accept" recommendation.

Reviewer #1: All comments have been addressed

2. Is the manuscript technically sound, and do the data support the conclusions?

Reviewer #1: Yes

3. Has the statistical analysis been performed appropriately and rigorously? 

Reviewer #1: Yes

4. Have the authors made all data underlying the findings in their manuscript fully available?

Reviewer #1: Yes

5. Is the manuscript presented in an intelligible fashion and written in standard English?

Reviewer #1: Yes

6. Review Comments to the Author

Reviewer #1: The authors responded to all of the questions/comments suggested. I feel that the manuscript is now suitable for further processing.

7. PLOS authors have the option to publish the peer review history of their article (what does this mean?). If published, this will include your full peer review and any attached files.

Reviewer #1: No

---

## [Author Response · Author response to Decision Letter 1]

13 Nov 2020

PONE-D-20-16107R1

Light promoted brown staining of protoplasm by Ag+ is ideal for testing wheat pollen viability

PLOS ONE

Response to Academic Editors Comments:

Firstly, thanks for email stating that you "feel that it has merit” (it probably refers to our manuscript)

Comment of Academic Editors: "It looks OK, but I have a few comments that need to be addressed by the authors before I can recommend the paper to be published.

Response: Firstly, thanks for stating "It looks OK". Suggestions/directions for annotations were made without giving valid reasons. Following are our comments to the suggestions/directions:

Comment of Academic Editors: "1) The author needs to revise the title"

Response: We are surprised to note that PLOS ONE’s publication criteria permits/encourages Academic Editors to direct authors to alter title of any manuscript at their own discretion without giving valid reasons.

However, we annotated the title of our manuscript from “Light promoted brown staining of protoplasm by Ag+ is ideal for testing wheat pollen viability” to “Light promoted brown staining of protoplasm by Ag+ is ideal to test wheat pollen viability rapidly”

Comment of Academic Editors: "2.) The data are mostly qualitative, therefore the authors need to tone down their claims about the novelty of the methodology."

Response: To the best of our knowledge all of the general cell viability or pollen viability test media or stains or protocols that are qualitative. We shall be highly obliged it you can kindly let us know any protocol that provide quantitative measurements through cell staining. Yes, one can furnish the data such as percentage of germination or percent of cells that remain viable after a specific treatment/stress. We don’t see any reason, why we can’t stress that our protocol is novel, when it is 100% novel? Please furnish us reasons or published literature where such a protocol was invented or even suggested. We never heard or aware of any means through which one can "tone down the claim about novelty of the methodology", in spite of being actually novel. We are surprised to note that PLOS ONE’s publication criteria permits Academic Editor to direct authors to tone down their claims about the novelty of the methodology, when the same is 100% novel.

Comment of Academic Editors: "3.) The authors need to carefully address the Reviewers comments in the manuscript (responding to the comments is not sufficient)."

Response: To the best of my knowledge, we completely addressed the Reviewers comments in the manuscript. Can you please let us know which one of the Reviewer's comments was not addressed by us in the manuscript? In fact, we completely restructured our manuscript in light of the Reviewer's and Academic Editor's comments. Accordingly, the Reviewer [who reviewed the revised version – i.e. PONE-D-20-16107R1] clearly stated/accepted that "All comments have been addressed" and "The authors responded to all of the questions/comments suggested. I feel that the manuscript is now suitable for further processing." Probably both the Academic Editors overlooked our response to the previous round of the Reviewer’s comments. We have specifically and pointwise stated at the end of the present round of the Comments of the Academic Editors and the Reviewers, how we annotated our manuscript taking into consideration each and every comment made by the Reviewers during first round of processing of our manuscript. 

Comment of Academic Editors: "4.) They authors should add more data to the paper. I edited many papers for PLoS One, but I never approved a paper with only one figure with mostly qualitative data"

Response: We failed to understand, what sort of additional data is desired. To the best of our knowledge, we carried out all the necessary and authentic investigations to furnish a novel simple protocol for testing wheat pollen viability. Our aim was to furnish a novel and simple protocol for testing wheat pollen viability and we furnished an authentic and undebatable protocol for testing wheat pollen viability. Further, our investigations also furnished necessary and authentic evidences to explain the mechanism associated with brown staining by Ag+ -viability medium. We unequivocally demonstrated that the brown staining is due to photoreduction of Ag+ to biphasic AgNPs. Do you find any lacunae in any of the most relevant methods used by us for characterization? Happy to know that you edited many papers for PLoS One, but I feel sad and surprised to know that you never approved a paper with only one figure with mostly qualitative data. To the best of my knowledge, an ideal Editor should focus and take decisions based on scientific merit and novelty rather than quantity of data and length of paper. As an Academic Editor my focus has always been towards scientific merit in terms of crispness and novelty along with unquestionable evidence(s). I am surprised to know that PLOS ONE’s publication criteria rejects manuscripts (i) with single figure or even no figure; (ii) whose length is short and crisp or when they have been written precisely. This comment of yours has given us an impression that PLOS ONE’s publication criteria permits only lengthy papers with lots of data.

However, for your satisfaction, we have split figure 1 and now manuscript has three figures instead of one. It certainly improved the clarity of information projected in the figures.

Comments of the Reviewer for PONE-D-20-16107R1:

Comment: "All comments have been addressed".

Response: Thanks for accepting and stating that we addressed all comments.

Comment: "The authors responded to all of the questions/comments suggested. I feel that the manuscript is now suitable for further processing."

Response: Thanks for further confirming that we responded to all of the questions/comments raised on the original version of our manuscript. Thanks also for emphasizing that our manuscript is suitable for further processing.

Response to the Original Version of the manuscript with comments for information to the new Academic Editors Dr. Chen and Dr. Joshi to stress that all relevant suggestions have been incorporated in the revised manuscript. As the manuscript has been totally restructured it is not possible to depict the same through track changes.

PONE-D-20-16107

Potential to generate Ag-nanoparticles can be used as an ideal tool to determine pollen viability

PLOS ONE

Academic Editor’s Comment: “After careful consideration, we have decided that your manuscript does not meet our criteria for publication and must therefore be rejected.”

Response: According to both the Reviewers 1 and 2, the manuscript was presented in an intelligible fashion and written in standard English. Reviewer 1 further stated that “The manuscript is written well and proper justifications are given to the results obtained.”

Further, all the issues raised by the reviewers are minor in nature and can be rectified with ease. To the best of our knowledge, none of the issues raised by the Reviewers are serious enough for “Rejection”. 

It is surprising that the Academic Editor, Dr. Vijay Kumar failed to give his own independent comments/views [he used only standard Rejection letter of PLoS One, without furnishing any Specific comments for the Rejection]. As none of the comments of the Reviewers are serious in nature, we believe that the “Rejection” decision taken by the Academic Editor, Dr. Vijay Kumar is 100% biased and it only reflects that he has some personal vengeance against us. 

While respecting the comments of the Reviewers, we annotated the manuscript significantly. 

As elaborating on the reasons for using each plant/crop species, would unnecessarily (i) dilute the key objective and findings of our investigations; and (ii) increase the length of the manuscript, we decided to restrict to only wheat [as detailed investigations were carried with this plant species] and also furnish data obtained using Ag+-viability medium consisting of AgNO3 and KNO3 [and totally omitted the information related with KCl]. In fact, the entire data furnished in the original manuscript was obtained using Ag+-viability medium consisting of AgNO3 and KNO3. Although, we obtained similar results even when we replaced KNO3 with KCl [as either of these salts are used as osmotic agents to prevent pollen bursting]. The same has been stated in our original manuscript. Unfortunately, due to negligence we mistakenly typed KCl in place of KNO3 while disclosing the composition of Ag+-viability medium. This mistake was committed as we wanted to disclose interesting observation, wherein we recorded staining of pollen brown by AgNO3 even in presence of KCl, although AgNO3+ KCl mixture turned colloidal due to formation of AgCl [we indeed specifically pointed out the formation of colloidal medium due to the formation of AgCl in our Original manuscript]. Probably the Academic Editor as well as the Reviewer 2 failed to go through our manuscript judiciously. To avoid any confusion, now we deleted KCl results. Please find below our comments on every point raised by both the Reviewers.

For Information of the Academic Editors Dr. Chen and Dr. Joshi: Accordingly, we have completely restructured the manuscript.

Reviewer #1 (R1)

R1 Comment: The manuscript entitled “Potential to generate Ag-nanoparticles can be used as an ideal tool to determine pollen viability” has been reviewed. “It is very general observation that plant cells and extracts could be used for biogenic synthesis of AgNPs.” 

“The manuscript is written well and proper justifications are given to the results obtained.”

Response: This Reviewer specifically stated/agreed that the manuscript is written well and proper justifications are given to the results obtained. We fail to understand how can a manuscript that is written well with proper justification to the findings/results can be blindly rejected by the Academic Editor Dr. Vijay Kumar. We accept that the biogenic synthesis of AgNPs by plant cells and extracts is a very general observation. However, objective of our investigations presented in the manuscript is not the general biogenic synthesis of AgNPs. Our aim was to develop a simple method for testing pollen viability. We used Ag+ for this endeavor and specifically demonstrated that the pollen viability can be tested using AgNO3. However, to explain the mechanism associated with brown staining, we specified that brown staining of pollen on incubation with AgNO3 was due to synthesis of AgNPs [which have been characterized for proving the mechanism associated with staining)]. 

For Information of the Academic Editors Dr. Chen and Dr. Joshi: This comment didn’t require any annotation in the manuscript.

R1 Comment: There are certain points to be rectified before further processing of the manuscript:

“• What was the criterion in selection of plants for pollen viability testing?”

“• Why these plants only selected?”

Response: Wheat is the major cereal crop [this information is now incorporated in the annotated manuscript] and continuous efforts are being made across the globe for its improvement through plant breeding program. Viable pollen is the key for successful plant breeding. 

Similarly, pigeon pea is an important pulse crop, Indian mustard is an important oil crop fennel is a spice crop, hemp is an important fiber crop, onion is an important vegetable crop. Black night shade is an important medicinal plant. Others are important floricultural crops, each of which possess some medicinal utility besides aesthetic value. Most importantly, all of them were easily available for our studies without any barrier. We don’t believe in increasing manuscript size by describing purpose of using each of them by briefing on importance of each floricultural plant species. In fact, such efforts will dilute the key objective of any useful investigation. We strongly believe that the manuscript should be crisp and focus on findings related to the objective/hypothesis, rather than unnecessarily increasing the length of the manuscript. Therefore, in annotated manuscript, we restricted ourselves to only wheat as we carried detailed and complete investigations with wheat pollen.

In fact, we submitted our manuscript as a Brief Communication to PLoS Biology and the same has been transferred to PLoS One [as an internal policy, after seeking permission of the Authors].

For Information of the Academic Editors Dr. Chen and Dr. Joshi: Accordingly, in annotated manuscript, we restricted ourselves to only wheat as we carried detailed and complete investigations with wheat pollen. [In fact, this Reviewer accepted the same and accordingly stated that all suggested corrections have been incorporated in the annotated manuscript].

R1 Comment: “• Part of second paragraph of Material and methods section should be presented in Results and discussion section.”

“• Even some parts of R and D section can be part of M and M section too.”

Response: Necessary action taken in Annotated manuscript.

For Information of the Academic Editors Dr. Chen and Dr. Joshi: In annotated manuscript, we had taken care of this issue [In fact, this Reviewer accepted that all suggested corrections have been incorporated in the annotated manuscript].

R1 Comment: “• What was the standard solution used for the spectroscopic analysis of AgNP containing pollens?”

Response: We used mixture containing 0.5 mM AgNO3 and 300 mM KNO3, which has been diluted by adding equal volume of distilled water, in a manner similar to the way viability medium incubated with pollen was diluted. The later has been stated clearly in our manuscript.

For Information of the Academic Editors Dr. Chen and Dr. Joshi: In annotated manuscript, we had taken care of this issue [In fact, this Reviewer accepted that all suggested corrections have been incorporated in the annotated manuscript]

R1 Comment: “• Write in details about the data analysis of this study.”

Response: We have already furnished information on data analysis. It is not clear what additional ‘details about the data analysis’ are required. However, we have further elaborated Methods.

For Information of the Academic Editors Dr. Chen and Dr. Joshi: In annotated manuscript, we had taken care of this suggestion [In fact, this Reviewer accepted that all suggested corrections have been incorporated in the annotated manuscript]

R1 Comment: “• Ag+ viability medium with wheat pollen grains turned brown within 2 min on incubation under ambient conditions in presence of solar radiation or high intensity visible light, but not in dark (Fig 1A). As evident from Fig 1B, protoplasm of pollen turned evenly brown on incubation in medium containing Ag+, only in presence of solar radiation or high intensity visible light. It is well established that solution containing Ag+ turns brown due to generation of AgNPs”....... The authors must discuss in details about the role of light in formation of AgNPs. Because, they got AgNPs in light conditions only.”

Response: At this stage, it is difficult for us to discuss details about the actual role of light in formation of AgNPs. We stated that high intensity light promoted rapid generation of AgNPs. However, if pollen are incubated under lab conditions for long duration, one can still see alteration in colour of viability medium with pollen, albeit the intensity of brown colour is low. Our objective is to develop a rapid pollen viability test. It is important to again stress that our aim was not to generate AgNPs. Generation of AgNPs by plant extracts has been carried out by innumerable number of investigators (surprisingly, even today such works get published) and in fact presently, generation of AgNPs by plant extracts has become a simple lab experiment for school and college students. At present, such works certainly don’t make any sense. 

For Information of the Academic Editors Dr. Chen and Dr. Joshi: In annotated manuscript, we elaborated on this suggestion [In fact, this Reviewer accepted that all suggested corrections have been incorporated in the annotated manuscript]

R1 Comment: “• Sentence should not be started with the abbreviations....”

Response: Suggested corrections incorporated.

For Information of the Academic Editors Dr. Chen and Dr. Joshi: In annotated manuscript, we had taken care of this suggestion [In fact, this Reviewer accepted that all suggested corrections have been incorporated in the annotated manuscript]

R1 Comment:”• It is well established fact that living cells and cellular organelles help in biogenesis of AgNPs, therefore, the pollen too helped in this study. Weather such pollens could be used for plant breeding experiments without losing their viability after used in this experiment?”

Response: To the best of our knowledge, use of silver salts for testing viability of pollen is novel. Yes, besides few other investigators, our research team has also authentically demonstrated that live cells as well as live chloroplasts possess potential to generate Ag-nanoparticles [some of which have been published in PLoS One itself]. In fact, this is the basic background (hypothesis) based on which we attempted to test if such a potential of live cells to generate Ag-nanoparticles can be exploited for testing pollen viability. It is relevant to cite few classical examples related for testing pollen viability itself. Earlier, few dynamic and pioneering researchers used TTC assay, esterase activity, peroxidase activity, IKI starch staining, acetocarmine test etc., all of which are known to exist in live cells for several decades. But, these dynamic researchers got due credit for using such approaches for testing pollen viability. We also wish to quote a dynamic pollen viability test published in PLOS One itself [Heidmann I, Schade-Kampmann G, Lambalk J, Ottiger M, Di Berardino M. Impedance flow cytometry: a novel technique in pollen analysis. PLoS One 2016;11: 0165531 doi: 10.1371/journal.pone.0165531], wherein Impedance flow cytometry was used for testing pollen viability. In fact, due to importance and relevance, researchers across the globe have been making efforts in developing novel and simple protocols for testing pollen viability. Similarly, recently Impe & co-workers published a paper on assessment of pollen viability in wheat in Frontiers in Plant science [Impe D, Reitz J, C Köpnick, Rolletschek H, Börner A, Senula A, Nagel M. Assessment of pollen viability for wheat. Front Plant Sci. 2020;10: 1588 doi: 10.3389/fpls.2019.01588]. Compared to the protocols used by these investigators as well as others so far, the pollen viability test developed by us is rapid, simplest and cost effective. 

For Information of the Academic Editors Dr. Chen and Dr. Joshi: In annotated manuscript, we addressed to this suggestion [In fact, this Reviewer accepted that all suggested corrections have been incorporated in the annotated manuscript]

R1 Comment: “• “in presence of solar radiation or high intensity light (Fig 1F & Fig 2L-2O). These findings, further establish that active/live protoplasm is necessary to turn pollen brown due to potential to generate AgNPs”.... But there are reports which used plant extracts only to generate AgNPs in the solution within one min.....any clarification in this regard...”

Response: Our aim was certainly not to furnish a protocol for generation of Ag-nanoparticles. Use of plant extracts for generation of Ag-nanoparticles has been known for several decades and presently, it is well known that such findings using blind use of extract of different parts of various plant species has no relevance as plant extracts possess innumerable types of reducing agents [such as phenolics, ascorbic acid, amino acids, organic acids, sugars, starch etc. etc.]. Accordingly, one gets highly heterogenous Ag-nanoparticles. If use of plants extracts has any significance, the same could have been used for commercial generation of Ag-nanoparticles. The same has been referred in our previous publications [including those appeared in PLoS One]. We could have given more specific comments, if this knowledgeable Reviewer could have pointed specific publications.

For Information of the Academic Editors Dr. Chen and Dr. Joshi: In annotated manuscript, we had taken care of this suggestion [In fact, this Reviewer accepted this response and accordingly stated that all suggested corrections have been addressed in the annotated manuscript]

R1 Comment: “• Please check “miccrocubes””

Response: Our aim is simple [i.e. furnishing a rapid and simple protocol for testing pollen viability]. We don’t find any relevance for testing “microcubes” [we don’t believe in blindly copying, what others have reported, until and unless it has relevance].

For Information of the Academic Editors Dr. Chen and Dr. Joshi: In annotated manuscript, we deleted information on microcubes due to lack of its relevance as we deleted information related to the use of KCl [In fact, this Reviewer accepted this response and accordingly stated that all suggested corrections have been addressed]

R1 Comment: “• References formatting must be uniform based on journal’s requirement.”

Response: This manuscript was initially submitted to PLoS Biology and the same has been transferred directly from PLos Biology website to PLoS One website. Yes, references have now been formatted in accordance with PLoS One’s requirement.

For Information of the Academic Editors Dr. Chen and Dr. Joshi: In annotated manuscript, we had taken care of this suggestion [In fact, this Reviewer accepted that all suggested corrections have been incorporated in the annotated manuscript]

R1 Comment: “• Authors should provide page and line numbers while revising the manuscript.”

Response: Page and line numbers have now been included.

For Information of the Academic Editors Dr. Chen and Dr. Joshi: In annotated manuscript, we had taken care of this suggestion [In fact, this Reviewer accepted that all suggested corrections have been incorporated in the annotated manuscript]

Reviewer #2 

R2 Comment: “The manuscript has some serious issues that need to be addressed:”

Response: Each of the “serious issues” raised by this Reviewer is addressed below.

R2 Comment:1. Experiment with a non-biological control (no pollen) wherein only Silver salt and KCl in the presence of light should be conducted.

Response: This was very much carried out, but inclusion of this information doesn’t add any significant addition to the manuscript. Further in annotated manuscript we deleted our findings with KCl.

For Information of the Academic Editors Dr. Chen and Dr. Joshi: In annotated manuscript, we deleted information related to KCl to avoid any sort of confusion.

R2 Comment:2. Experiment with a non-biological control (no pollen) wherein only Silver salt and KCl in the absence of light should also be conducted. Please refer to the paper wherein SNPs (AgNPs) and GNPs were generated with heat-killed date palm pollens in the dark.

Response: This was also very much carried out and we did categorically state in our manuscript that “on mixing AgNO3 with KCl, an opaque white colloidal solution was produced as a result of formation of AgCl within a short duration”. Therefore, we used KNO3 and not KCl, although we do get similar results with KCl. The same has been stated in Materials and methods. Unfortunately, we have mistakenly referred KNO3 as KCl. We apologize for the same. 

We have carefully gone through the paper of Banu et al. (2018) referred by this reviewer and noted that (i) heat-killed date palm pollen were not directly used for generation of Ag and Au nanoparticles; (ii) the extract obtained by heating stored pollen dust, was used for generating nanoparticles in dark; and (iii) investigations were carried with the presumption that the extract contains a range of phytochemicals. No attempts were made by these authors to evaluate if pollen themselves can generate nanoparticles. Hundreds of papers have been published where researchers demonstrated potential of extracts of plant parts of various plant species to generate nanoparticles. We don’t see any reason, why we need to cite this paper as (i) pollen were not directly used for generation of nanoparticles; and (ii) the authors showed photos of dried inflorescence of date palm as a figure and not of pollen. Further, the findings presented in Banu et al. (2018) are nothing to do with pollen viability or any way closure to the hypothesis based on which we performed our investigations presented in our manuscript. As stated previously, we wish to again stress that our aim was not to generate Ag-nanoparticles. 

For Information of the Academic Editors Dr. Chen and Dr. Joshi: Due to lack of any relevance, we didn’t incorporate this suggestion in annotated manuscript.

R2 Comment: “3. There are ample reports describing synthesis of SNPs using whole cell, cell free extract, plant extract for synthesis of nanoparticles using Ag salts only. I would like to know why KCl was used in the reaction as it is well known that Ag salt will be precipitated as Ag chloride.”

Response: Our research team itself has reported potential of chloroplasts and root cell (in particular) epidermal cells generate Ag-nanoparticles. Use of plant extracts to generate AgNPs is known from last three decades. Objective of our investigations is not to develop any AgNPs generating protocol. We specifically stated that our objective was to test if rapid AgNPs synthesis capacity of live pollen can be exploited for testing pollen viability.

In our original manuscript, we had categorically stated that “on mixing AgNO3 with KCl, an opaque white colloidal solution was produced as a result of formation of AgCl within a short duration”. This itself reflects that we are very well aware that Ag+ precipitates as AgCl. That is the precise reason, why we stated in Materials and Methods “Therefore, it is important to disclose specifically that 1 mM AgNO3 and 600 mM KCl solutions must be stored separately and former solution be added soon after pollen were incubated in later solution.” Although, we were getting ideal results with KNO3 (which we used for all our investigations presented in the manuscript), we gave this information as we were getting equally good results with KCl. However, through the above referred statement, we wanted researchers (who prefer using KCl) to know that they should incubate pollen first in KCl and then add AgNO3. This allows K+ and Cl- to be absorbed by the pollen grains before Ag+. Moreover, the concentration of Ag+ used was 600 times lower than that of KCl. Further, it is important to disclose true facts to the scientific community. If we wanted, we could have avoided disclosing results we obtained with KCl. To avoid any sort of confusion, now we deleted the findings with KCl in annotated manuscript.

For Information of the Academic Editors Dr. Chen and Dr. Joshi: To avoid any sort of confusion, we deleted the findings with KCl in annotated manuscript.

R2 Comment: “4. Experiment using Ag salts and pollen may also be conducted.”

Response: We have specifically stated in our manuscript [based on authentic reports from several researchers, besides common observation made by even school kids studying pollen germination/biology], the significance of using an osmotic component to prevent bursting of pollen. We also stated categorically that, we diluted viability medium with pollen incubated in it, to burst the pollen so that the contents get released due to bursting. Therefore, performing any investigations of pollen viability just with silver salt is unscientific and irrational effort as live pollen burst due to severe osmotic shock (extreme hypotonic condition). However, if someone is interested in generating Ag-nanoparticles, such an effort would be ideal. 

For Information of the Academic Editors Dr. Chen and Dr. Joshi: Due to lack of any relevance, we didn’t incorporate this suggestion in annotated manuscript. Dr. Joshi must be very clear with concepts related to osmotic components.

R2 Comment: “5. Authors have not divulged details on various accounts, e.g. sonication conditions, heat-killing treatment and aged pollen.”

Response: As stated previously, we submitted this manuscript to PLoS Biology as a Brief Communication and hence didn’t elaborate on “sonication conditions, heat-killing treatment and aged pollen”. Now, we included the same in our annotated manuscript.

For Information of the Academic Editors Dr. Chen and Dr. Joshi: These suggestions have been incorporated in annotated manuscript.

R2 Comment:6. Please see the annotated pdf for detailed comments.

Suggestion of the Reviewer 2: “Subsequent to pollen rupturing” instead of “Subsequent to rupturing pollen”.

Response: We don’t accept this suggestion as it gives a wrong impression.

In addition, more or less similar comments as above were marked on the PDF. The same have been extracted in JPG format and pasted below along with our specific response:

For Information of the Academic Editors Dr. Chen and Dr. Joshi: Due to lack of any relevance, we didn’t incorporate this suggestion in annotated manuscript.

R2 Comment on PDF:

Response: It is well known/established fact that viability of pollen varies from species to species. Pollen tend to lose viability on storage, especially under ambient conditions. The term ”aged” was used by us for the pollen stored under ambient conditions for different time intervals. To avoid confusion, now we deleted the term “aged” and stated the duration of storage in the annotated manuscript. 

For Information of the Academic Editors Dr. Chen and Dr. Joshi: Due to lack of any relevance, we didn’t incorporate this suggestion in annotated manuscript.

R2 Comment on PDF:

Response: I don’t see any valid reason why we need to unnecessarily add a Table to increase the length of the manuscript. There is absolutely nothing wrong in simply listing the plant species used. However, authority name has now been included. It is irrelevant to refer to www.ipni.org as there are superior sites. 

The question of “where are the vouchers of the species used deposited?”, only reflects negative mindset of the Reviewer [as well as the Academic Editor Dr. Vijay Kumar”, who entertained such incompetent comments]. Can Dr. Vijay Kumar let me know what sort of vouchers this Reviewer is referring to? Does every author who is using a plant species authorized to submit such vouchers? 

With regard to the reason, why we selected different plant species, we made it crystal clear in our original manuscript that “In order to check, if Ag+ viability medium can be used for testing pollen viability in all plant species, we extended our studies to 20 plant species …..” besides their availability.

For Information of the Academic Editors Dr. Chen and Dr. Joshi: In annotated manuscript, we restricted our findings to only wheat, as we carried detailed investigations with wheat pollen.

R2 Comment on PDF:

Response: That is the precise reason, why we categorically stated the same in our manuscript. We disagree that it is essential to use SEM, AFM to confirm the presence of AgNPs. I am shocked. It is not clear how AFM confirms the presence of AgNPs. To the best of our knowledge Transmission electron microscope coupled with the hardware for (a) energy dispersive X-ray (EDX) analysis; and (b) selected area electron diffraction (SAED) pattern analysis, is good enough to characterize AgNPs. In addition, we carried out vital PXRD analysis. 

For Information of the Academic Editors Dr. Chen and Dr. Joshi: Due to lack of any relevance, we didn’t carry any additional investigations with AFM and SEM as we obtained authentic and undebatable information through TEM and PXRD investigations. 

R2 Comment on PDF:

Response: We have carefully gone through the paper of Banu et al. (2018) referred by this reviewer and noted that (i) heat-killed date palm pollen were not directly used for generation of Ag and Au nanoparticles; (ii) the extract obtained by heating stored pollen dust, was used for generating nanoparticles in dark; and (iii) investigations were carried with the presumption that the extract contains a range of phytochemicals. No attempts were made by these authors to evaluate if pollen themselves can generate nanoparticles. The authors showed photos of dried inflorescence of date palm as a figure and not of pollen. Further, the findings presented in Banu et al. (2018) are nothing to do with pollen viability or any way closure to the hypothesis based on which we performed our investigations presented in our manuscript. As stated previously, we wish to again stress that our aim was not to generate Ag-nanoparticles. Unfortunately, this expert reviewer seems to be interested only in generation of Ag-nanoparticles, that too using extracts of plants parts. Our research team never believed in blindly using extracts [which contain wide range of biomolecules with different degrees of reducing capacity] for generating metal nanoparticles.

For Information of the Academic Editors Dr. Chen and Dr. Joshi: Due to lack of any relevance, we didn’t incorporate this suggestion in annotated manuscript.

R2 Comment on PDF:

Original: “grasped”

Reviewer 2’s suggestion: “captured”

Response: This suggestion has been incorporated in the annotated manuscript.

For Information of the Academic Editors Dr. Chen and Dr. Joshi: This suggestion has been incorporated in annotated manuscript.

R2 Comment on PDF:

Response: Sonication conditions have been included in the annotated manuscript.

For Information of the Academic Editors Dr. Chen and Dr. Joshi: This suggestion has been incorporated in annotated manuscript.

R2 Comment on PDF:

Response: Although we carried such control experiments, we didn’t feel any necessity to include the same as inclusion of such results don’t add any relevance to the basic objective, hypothesis and the novel findings of the investigations presented in our manuscript. Moreover, in the annotated manuscript we deleted the findings that we obtained with KCl as we used KNO3 constantly for testing pollen viability.

For Information of the Academic Editors Dr. Chen and Dr. Joshi: To avoid any sort of confusion as stated previously, we deleted the findings with KCl in annotated manuscript.

R2 Comment on PDF:

Response: Details of how pollen was heat killed has now been elaborated in the annotated manuscript.

For Information of the Academic Editors Dr. Chen and Dr. Joshi: This suggestion has been incorporated in annotated manuscript.

R2 Comment on PDF: Names of all Journals have been highlighted as they were represented in full form rather than in abbreviated form. 

Response: We have now abbreviated the names of the journals in the annotated manuscript.

For Information of the Academic Editors Dr. Chen and Dr. Joshi: This suggestion has been incorporated in annotated manuscript.

---

## [Editor Report · Decision Letter 2]

30 Nov 2020

Light promoted brown staining of protoplasm by Ag+ is ideal to test wheat pollen viability rapidly

PONE-D-20-16107R2

Dear Dr. Pardha-Saradhi,

We’re pleased to inform you that your manuscript has been judged scientifically suitable for publication and will be formally accepted for publication once it meets all outstanding technical requirements.

Kind regards,

Zhong-Hua Chen, Ph.D.

Academic Editor

PLOS ONE

Additional Editor Comments (optional):

The Revision is acceptable now.
---

## [Editor Report · Acceptance letter]

2 Dec 2020

PONE-D-20-16107R2 

Light promoted brown staining of protoplasm by Ag^+^ is ideal to test wheat pollen viability rapidly 

Dear Dr. Pardha-Saradhi:

I'm pleased to inform you that your manuscript has been deemed suitable for publication in PLOS ONE. Congratulations! Your manuscript is now with our production department. 

Kind regards, 

on behalf of

Dr. Zhong-Hua Chen 

Academic Editor

PLOS ONE